# Endoscopic Retrograde Cholangiopancreatography (ERCP) for Suspected Mirizzi Syndrome Type IV as Both a Diagnostic and Bridge-to-Surgery Procedure

**DOI:** 10.3390/diagnostics14080855

**Published:** 2024-04-22

**Authors:** Giacomo Emanuele Maria Rizzo, Settimo Caruso, Ilaria Tarantino

**Affiliations:** 1Endoscopy Service, Department of Diagnostic and Therapeutic Services, IRCCS-ISMETT, 90127 Palermo, Italy; 2Department of Precision Medicine in Medical, Surgical and Critical Care (Me. Pre. C. C.), University of Palermo, 90127 Palermo, Italy; 3Radiology Service, Department of Diagnostic and Therapeutic Services, IRCCS-ISMETT, 90127 Palermo, Italy

**Keywords:** Mirizzi syndrome, surgery, endoscopy, biliary tract, ERCP

## Abstract

Mirizzi syndrome (MS) is a challenging diagnosis due to its similar presentation with other biliary diseases; thus, the role of endoscopy is sometimes unclear, especially in altered anatomy. Radiological examinations may usually suspect it, but deeper examinations could be necessary to confirm it. Endoscopic retrograde cholangiopancreatography (ERCP) certainly has a therapeutic role in cases of jaundice, cholangitis or concurrent choledocolithiasis, although surgery is without doubt the definitive treatment in most of the cases. Therefore, surgeons may have a clearer picture of the condition of the biliary tree with respect to fistulas thanks to ERCP, particularly in patients with a higher grade of MS (type higher than 2 in the Csendes classification). Therefore, a complete removal of biliary stones is sometimes not possible due to size and location, so biliary stenting becomes the only option, even if transitory. Our brief report is a further demonstration of the fundamental role of ERCP in managing MS, even when it has no long-term therapeutic aim but is performed as bridge-to-surgery, especially in cases with a more difficult biliary anatomy due to the type of fistula. Moreover, we truly suggest discussing patients affected with MS in a multidisciplinary board, preferably in tertiary hepatobiliary centers.

**Figure 1 diagnostics-14-00855-f001:**
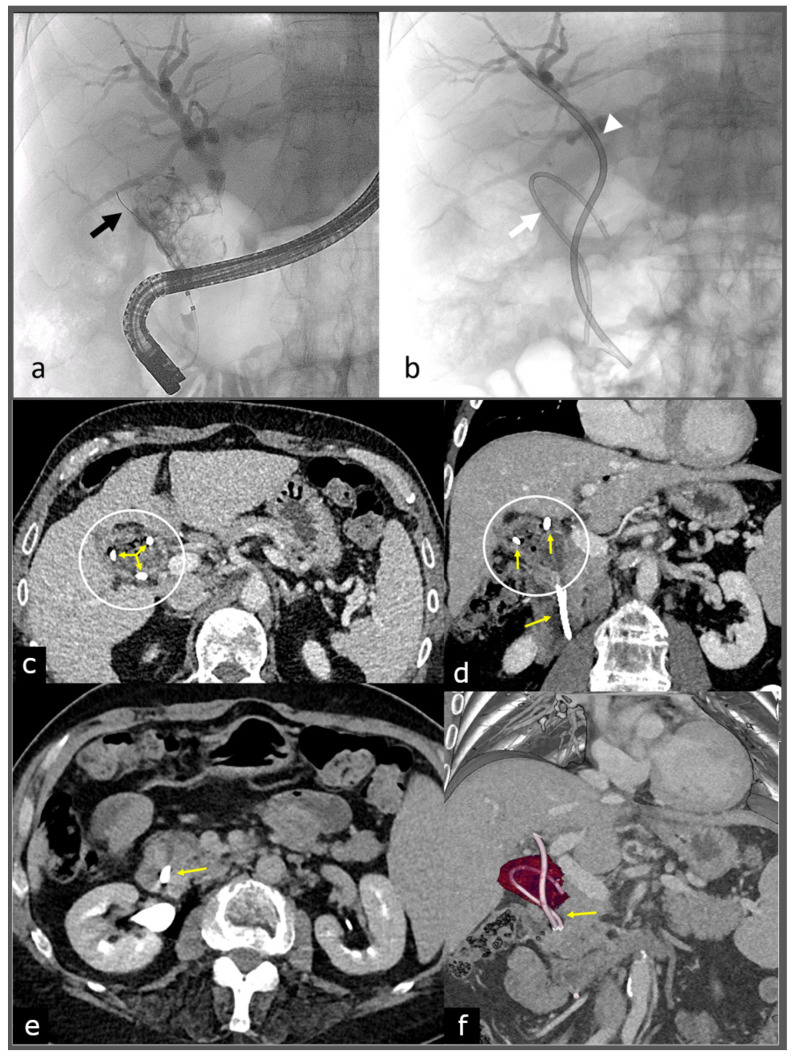
(**a**) Cholangiography during endoscopic retrograde cholangiopancreatography (ERCP) shows common bile duct (CBD) enlarged and full of biliary stones, next to a large, irregular and sack-like area full of stones adhering and undistinguished to the middle third of CBD, apparently corresponding to the gallbladder, giving an initial suspect of Mirizzi Syndrome (MS, with a large cholecysto-choledochal fistula). MS is a challenging diagnosis due to its similar presentation with other biliary diseases; thus, the role of endoscopy is sometimes unclear, especially in altered anatomy [1,2,3]. Guide for stent placement (black arrow). This particular case is a 62-year-old male who had a sudden onset of abdominal pain and jaundice. He had history of asymptomatic gallbladder stones, so magnetic resonance imaging (MRI) was performed after the onset of the symptoms showing common bile duct (CBD) and gallbladder lithiasis with biliary duct dilation. (**b**) Biliary drainage with two biliary plastic stents (yellow arrows) was preferred over other alternatives, such as endoscopic nasobiliary drainage (ENBD), due to the higher risk of catheter dislocation in case of ENBD if the patient would retract or remove it. The thinner one of 7 Fr (white arrow) creates a loop surrounding the “gallbladder” edge, while the thicker one of 10 Fr (head of arrow) passes through biliary stones to the intra-hepatic dilated ducts in order to permit complete biliary drainage. (**c**,**d**) CT axial (**c**) and coronal (**d**) images show a large duct of 40 mm with hypodense (no calcified gallstones) and inhomogeneous content (white circle). Gallbladder and main biliary duct are no longer individually recognizable, but only thanks to the two stents. (**e**) CT axial image shows no dilatation of intrapancreatic CBD. (**f**) Multi intensity projection (MIP) and volume rendering construction CT image showing, in purple, the dilated CBD due to the entire fistulation of the gallbladder. These findings seemed to confirm the diagnosis of MS type IV (according to Csendes classification) [4].

**Figure 2 diagnostics-14-00855-f002:**
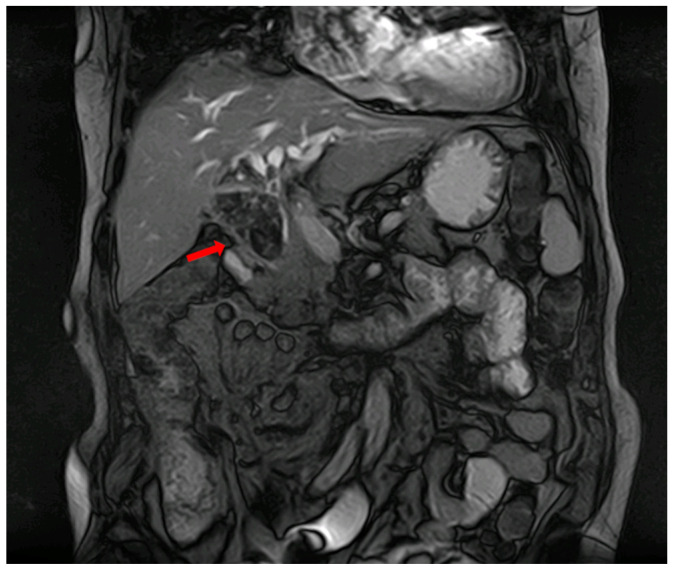
MRI showing common bile duct (CBD) and gallbladder lithiasis (red arrow) associated with biliary duct dilation.

**Figure 3 diagnostics-14-00855-f003:**
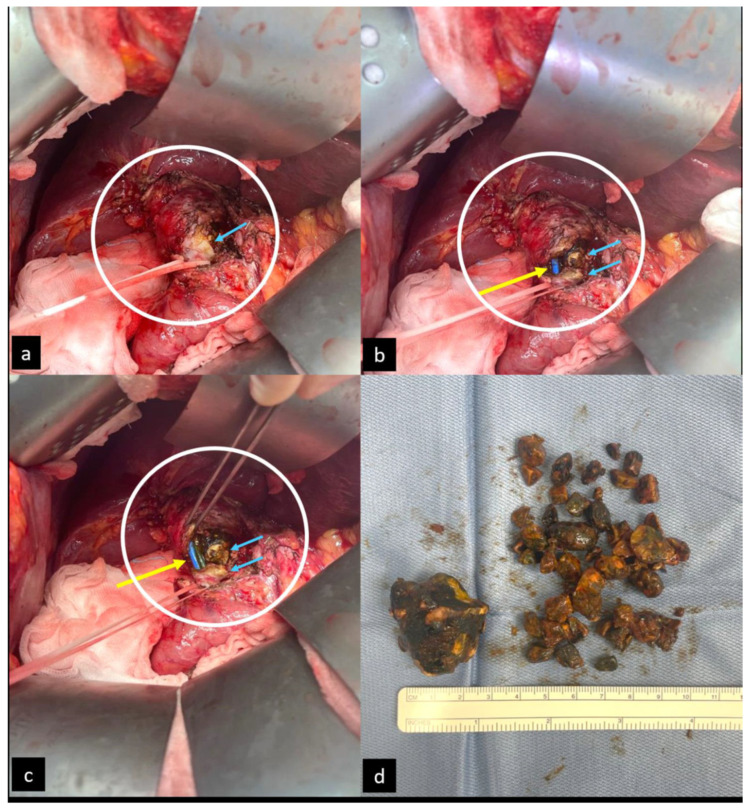
The patient was clinically stable, and the bilirubin was dropping to normal serological levels; therefore, elective surgery was scheduled after a careful analysis during a multidisciplinary meeting. Therefore, surgery is without doubt the definitive treatment in most of the cases [5,6]. Pictures (**a**–**c**) illustrate the suprapancreatic large main biliary tract that was partially resected during surgery (white circle). The large biliary stone (blue arrows) and biliary plastic stents emerge from the main biliary duct (yellow arrow) after its initial incision where the large cholecystocholedochal pseudofistula replaces the normal route of the main biliary tract. A bilio-enteric diversion was promoted thanks to the location of the stents, which worked as landmarks to guide the surgical technique. (**d**) The various sizes of the biliary stones removed from the biliary tract at the end of the surgery. The patient was discharged after one week without any complications, and he did not complain about any biliary signs or symptoms after 6 months of follow up.

## Data Availability

Data available on request due to restrictions (e.g., privacy, legal or ethical reasons).

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
