# Peer review of "Endoscopic Retrograde Cholangiopancreatography (ERCP) for Suspected Mirizzi Syndrome Type IV as Both a Diagnostic and Bridge-to-Surgery Procedure"

_diagnostics, 2024, doi:10.3390/diagnostics14080855_

Round 1

Reviewer 1 Report

Comments and Suggestions for Authors

ERCP is an important part of treating MS and pre-operational ERCP with stenting or ENBD might decrease the rate of peri-operative complications. Though MS can sometimes be difficult to diagnose, MRCP might provide clues to the diagnosis and classification of MS. Thus, it would be better if the image and description of the patient's MRCP be shown to the readers. And the reason that you choose to drain the patient with 2 plastic stents instead of other strategies like ENBD could also be explained to the readers and discussed in the manuscript. 

Author Response

Thank you for your appreciation and suggestions. We added the figure of the MRCP and explanation. We didn’t add many explanations about biliary drainage strategy on the paper due to the editorial limits for this type of article. However, we preferred to put two plastic stents to obtain a more stable internal drainage, reducing the possibility to dislocation. In fact, the ENBD may be an alternative option, but we did not use it because the patients sometimes try to remove or retract it, increasing the possibility to dislocate or remove it. We added reasons and explanations in the text of the manuscript.

Reviewer 2 Report

Comments and Suggestions for Authors I think it is a good article and with an interesting case. Yes, accept the manuscript for publication.

Author Response

Thank you very much for your appreciation.